# GROUNDING MULTIMODAL LARGE LANGUAGE MODELS TO THE WORLD

**Zhiliang Peng**[1*]  **Wenhui Wang**[2*]  **Li Dong**[2*]
**Yaru Hao**[2]  **Shaohan Huang**[2]  **Shuming Ma**[2]  **Qixiang Ye**[1†]  **Furu Wei**[2†]
[1]University of Chinese Academy of Sciences
[2]Microsoft Research
https://aka.ms/GeneralAI

## ABSTRACT

We introduce KOSMOS-2, a Multimodal Large Language Model (MLLM), enabling new capabilities of perceiving object descriptions (*e.g.*, bounding boxes) and grounding text to the visual world. Specifically, we represent text spans (*i.e.*, referring expressions and noun phrases) as links in Markdown, *i.e.*, "[text span](bounding boxes)", where object descriptions are sequences of location tokens. To train the model, we construct a large-scale dataset about grounded image-text pairs (GRIT) together with multimodal corpora. KOSMOS-2 integrates the grounding capability to downstream applications, while maintaining the conventional capabilities of MLLMs (*e.g.*, perceiving general modalities, following instructions, and performing in-context learning). KOSMOS-2 is evaluated on a wide range of tasks, including (i) multimodal grounding, such as referring expression comprehension and phrase grounding, (ii) multimodal referring, such as referring expression generation, (iii) perception-language tasks, and (iv) language understanding and generation. This study sheds a light on the big convergence of language, multimodal perception, and world modeling, which is a key step toward artificial general intelligence. Code can be found in https://aka.ms/kosmos-2.

## 1 INTRODUCTION

Multimodal Large Language Models (MLLMs) (Hao et al., 2022; Alayrac et al., 2022; Huang et al., 2023; Driess et al., 2023; OpenAI, 2023) have been a general-purpose interface across language, vision, and vision-language tasks. They are able to perceive general modalities, including texts, images, and audio, and generate responses using free-form texts under zero-shot and few-shot settings.

In this study, we unlock the referring and grounding capabilities of multimodal large language models, with the aim to construct a more flexible and general human-computer interface about vision-language tasks, Figure 1. With such a model, users can directly point to objects or image regions without requiring detailed text descriptions referring to them. It also enables the model to respond with visual answers (*i.e.*, bounding boxes), supporting more vision-language tasks such as referring expression comprehension while resolving their co-reference ambiguity.

Our proposed model, referred to as KOSMOS-2, is a Transformer-based causal language model built upon KOSMOS-1 (Huang et al., 2023), but has the major differences of grounding and referring capabilities. To unlock the grounding capability, we first construct a web-scale dataset of grounded image-text pairs (GRIT), which are built upon a subset of image-text pairs from LAION-2B (Schuhmann et al., 2022) and COYO-700M (Byeon et al., 2022). GRIT is combined with the multimodal corpora (*i.e.*, text corpora, image-text pairs and interleaved image-text data) to train the model. To construct GRIT, we propose an approach to extract and link text spans (*i.e.*, noun phrases and referring expressions) in image captions to spatial coordinates (*e.g.*, bounding boxes) of the corresponding objects or image regions. Spatial coordinates of each object are converted to a sequence of location tokens, which are appended atop text span of the object as an expansion. The expanded text span serves as a "*hyperlink*" ([text span](location tokens)) to connect the objects or regions of the image to the caption, as shown in Figure 1. Given the "*hyperlink*" data and trained in the

---

* Equal contribution. † Corresponding authors.

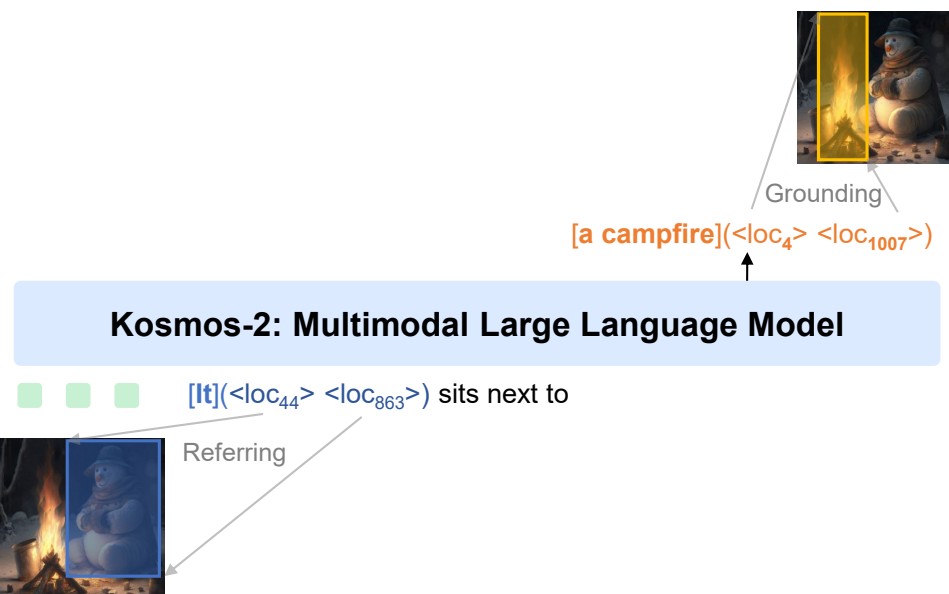

Figure 1: KOSMOS-2 is a multimodal large language model which has new capabilities of multimodal grounding and referring. KOSMOS-2 can understand multimodal input, follow instructions, perceive object descriptions (*e.g.*, bounding boxes), and ground language to the visual world.

causal language modeling fashion, KOSMOS-2 links text spans in the generated free-form text response to image regions, which facilities generating more accurate, informational, and comprehensive vision-language predictions. Utilizing a pronoun as the text span in conjunction with location tokens, KOSMOS-2 can perceive the referring object and incorporate it into the reasoning process. This simple transform enables the ability of referring, providing a more flexible interaction interface.

Experimental results show that KOSMOS-2 achieves not only competitive performance on language and vision-language tasks, but also leading performance on grounding tasks (phrase grounding and referring expression comprehension) and referring tasks (referring expression generation). The grounding capability born with KOSMOS-2 enables it be applied to more downstream tasks, such as grounded image captioning, and grounded visual question answering.

## 2 GRIT:WEB-SCALE GROUNDED IMAGE-TEXT PAIRS

To learn the grounding capability, we first construct a large-scale dataset of **Gr**ounded **I**mage-**T**ext pairs (GRIT), based on image-text pairs from a subset of COYO-700M (Byeon et al., 2022) and LAION-2B (Schuhmann et al., 2022). To this end, a pipeline is designed to extract and link text spans (*i.e.*, noun phrases and referring expressions) in the caption to their corresponding image regions, Figure 2. The pipeline consists of two steps: generating noun-chunk-bounding-box pairs and producing referring-expression-bounding-box pairs, which are detailed in what follows.

**Step-1: Generating noun-chunk-bounding-box pairs**   Given an image-text pair, we first extract noun chunks from the caption and associate them with image regions through a pretrained object detector. In specific, spaCy (Honnibal et al., 2020) is employed to parse the caption ("*a dog in a field of flowers*") and extract all noun chunks ("*a dog*", "*a field*" and "*flowers*"). To reduce potential noise, we eliminate certain abstract noun phrases that are challenging to recognize in the image, such as "*time*", "*love*", and "*freedom*". Subsequently, the image and extracted noun chunks are fed to a pretrained grounding model (*e.g.*, GLIP (Li et al., 2022b)) to obtain the associated bounding boxes. The non-maximum suppression algorithm is applied to remove bounding boxes that have a high overlap with others, regardless of whether they are associated with the same noun chunk or not. The noun-chunk-bounding-box pairs with predicted confidence scores higher than 0.65 are kept. If no bounding boxes are retained, we discard the corresponding image-caption pair.

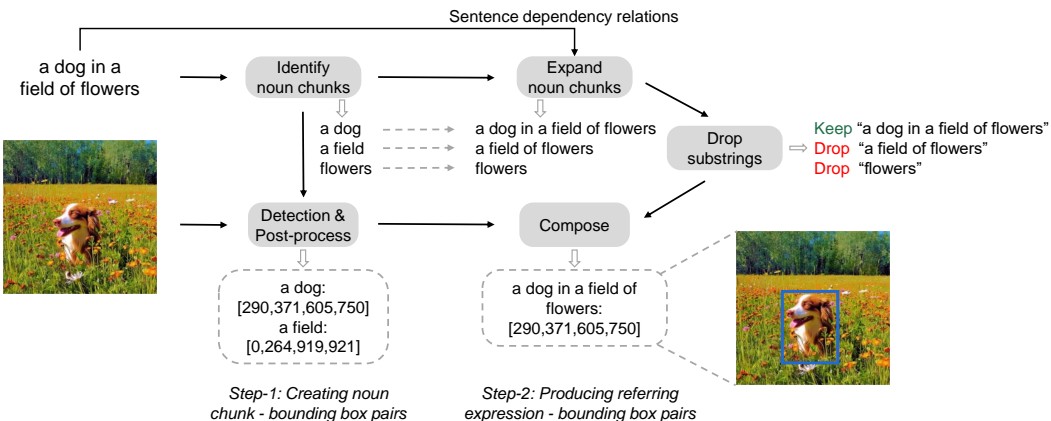

Figure 2: The pipeline to construct web-scale grounded image-text pairs.

**Step-2: Producing referring-expression-bounding-box pairs** To endow model with the ability to ground diverse and complex linguistic descriptions, we expand noun chunks to referring expressions. In specific, spaCy is employed again to obtain dependency relations of the caption. A noun chunk is then expanded to a referring expression by recursively traversing its children in the dependency tree and concatenating children tokens with the noun chunk. We do not expand noun chunks with conjuncts. The noun chunks without children tokens are retained for the subsequent process. As illustrated in Figure 2, the noun chunk 'a dog' is expanded to "*a dog in a field of flowers*", the noun chunk 'a field' is expanded to "*a field of flowers*", and the noun chunk '*flowers*' remains unchanged as it cannot be expanded.

Furthermore, we only retain text spans that are not contained by others. As demonstrated in Figure 2, we keep the referring expression "*a dog in a field of flowers*" and drop "*a field of flowers*" and '*flowers*' (as they are contained by "*a dog in a field of flowers*"). The bounding box of the noun chunk ('*a dog*') is assigned to the corresponding generated referring expression ("*a dog in a field of flowers*").

So far, we have more than 91M images, 115M text spans, and 137M associated bounding boxes. In comparison with publicly available visual grounding datasets, as shown in Table 8 in Appendix C.1, GRIT significantly improves the data scale. More data samples of GRIT are shown in Appendix C.3.

## 3 KOSMOS-2: GROUNDED MULTIMODAL LARGE LANGUAGE MODEL

KOSMOS-2 is a grounded MLLM, which follows the model architecture and training objective of KOSMOS-1, but integrates grounding and referring capabilities. For example, a KOSMOS-2 model can accept image regions drawn by users, provide visual answers (*i.e.*, bounding boxes), and ground the text output to the visual world. To endow the model with grounding and referring capabilities, we add grounded image-text pairs to the training data. For a text span (*e.g.*, noun phrase and referring expression) and its corresponding bounding boxes in a grounded image-text pair, we discretize continuous coordinates of bounding boxes to a sequence of location tokens which are uniformly encoded alongside text tokens. We then link the location tokens and their corresponding text span via a "*hyperlink*" data format. Based on grounded image-text pairs, a KOSMOS-2 model is trained to establish a mapping between image regions and their corresponding location tokens and connect the image regions with their associated text spans.

### 3.1 GROUNDED INPUT REPRESENTATION

Given a text span and its associated bounding boxes in a grounded image-text pair, we first convert the continuous coordinates of bounding boxes to a sequence of discrete location tokens (Chen et al., 2021). For an image with width $W$ and height $H$, we respectively divide both the width and height into $P$ segments. A total of $P \times P$ bins are calculated, with each bin comprising $(W/P) \times (H/P)$ pixels. For each bin, we use a location token to represent the coordinates within that bin. When

mapping location tokens back to the continuous coordinates in the image, we utilize the center pixel coordinates of each bin.

Each bounding box can be represented using its top-left point $(x_{tl}, y_{tl})$ and bottom-right point $(x_{br}, y_{br})$, which are discretized to top-left and bottom-right location tokens, respectively. A top-left location token (`<loc`$_{tl}$`>`) and a bottom-right one (`<loc`$_{br}$`>`) are used with special boundary tokens (`<box>` and `</box>`) to form the bounding box representation: `<box><loc`$_{tl}$`><loc`$_{br}$`></box>`. In instances where the text span is linked to multiple bounding boxes, a special token, `<delim>`, is employed to concatenate the location tokens of these bounding boxes. The resulting representation can be expressed as: `<box><loc`$_{tl}^i$`><loc`$_{br}^i$`><delim>...<loc`$_{tl}^j$`><loc`$_{br}^j$`></box>`.

Accordingly, we arrange the text span and its associated location tokens in a format resembling a "*hyperlink*" in Markdown. For the text span with a single bounding box, the resultant sequence is "`<p>` *text span* `</p><box><loc`$_{tl}$`><loc`$_{br}$`></box>`", where `<p>` and `</p>` are special tokens indicating the beginning and end of the text span. Such a format conveys to the model that image regions within `<loc`$_{tl}$`><loc`$_{br}$`>` are associated with '*text span*'.

Take Figure 1 as an exmaple, the input representation is:

> `<image>`Image Embedding `</image><grounding><p>`It`</p><box><loc`$_{44}$`><loc`$_{863}$`>` `</box>` sits next to `<p>`a campfire`</p><box><loc`$_{4}$`><loc`$_{1007}$`></box>`

where `` and `` indicate the start- and end-of-sequence, and `<image>` and `</image>` represent the beginning and end of encoded image embeddings. `<grounding>` is a special token employed to signal the model that the subsequent sequence includes text spans and their associated location tokens. We map input text tokens and location tokens to embeddings via a lookup table. A vision encoder, accompanied by a re-sampler module, is utilized to acquire image embeddings.

For language-only data, cross-modal paired data (*i.e.*, image-text pairs), and interleaved multi-modal data, we use the same input representations as of KOSMOS-1. Therefore, in these cases, the `<grounding>` token is not required to be prepended.

## 3.2 GROUNDED MULTIMODAL LARGE LANGUAGE MODEL

KOSMOS-2 uses a Transformer-based causal language model as the foundation architecture and is trained through the autoregressive language modeling task. In addition to multimodal corpora used in KOSMOS-1 (including text corpora, image-caption pairs, and interleaved image-text data), we add GRIT into training. The training loss only considers discrete tokens, such as text and location tokens. The model learns to locate and understand image regions through the location tokens, associate text spans to image regions, and output bounding boxes of the image region using location tokens.

KOSMOS-2 enhances MLLMs (Huang et al., 2023) by incorporating grounding and referring capabilities. Specifically, we can use "`<grounding>`...`<p>`*text span*`</p>`" as input to prompt KOSMOS-2 to generate location tokens for the '*text span*' in multimodal grounding tasks. We can also employ a pronoun as a text span in conjunction with location tokens, such as "`<grounding>`...`<p>`*It* `</p><box><loc`$_{tl}$`><loc`$_{br}$`></box>`", to enable KOSMOS-2 to perceive the referring objects or regions, providing a flexible human-computer interaction fashion. Furthermore, we can simply prepend the '`<grounding>`' token in conventional vision-language tasks (like image captioning) to facilitate new applications, resulting in more accurate, informative, and comprehensive responses.

## 3.3 MODEL TRAINING

**Setup** KOSMOS-2 is trained upon GRIT, text corpora, image-caption pairs, and interleaved image-text data. The training procedure involves a batch size of 419K tokens, consisting of 185K tokens from text corpora, 215K tokens from original and grounded image-caption pairs, and 19K tokens from interleaved data. The model is trained for 60K steps, utilizing approximately 25 billion tokens, using an AdamW optimizer with $\beta = (0.9, 0.98)$, a weight decay of 0.01, and a dropout rate of 0.1. The learning rate increases to 2e-4 during the first 375 warm-up steps and linearly decays to zero. The model is trained on 256 V100 GPUs for 24 hours. To tell the model when to ground text output to the visual world, we prepend the '`<grounding>`' token to the grounded caption during training.

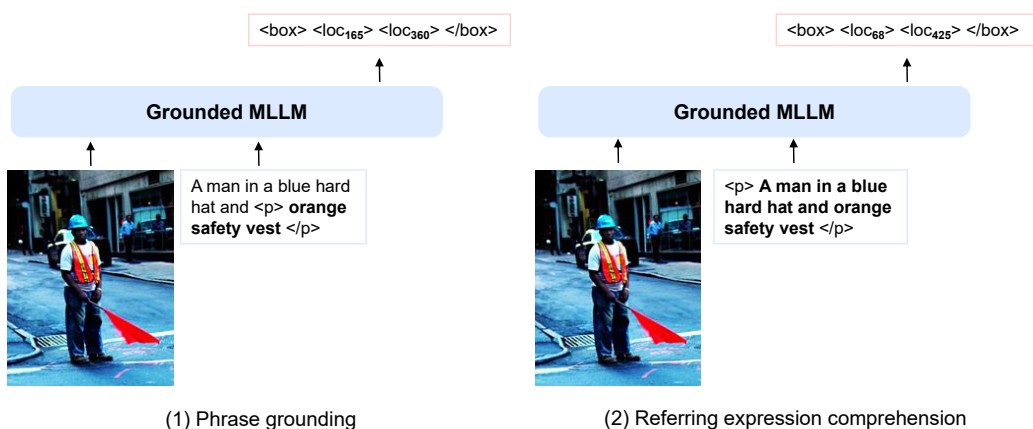

Figure 3: Input format of evaluations on phrase grounding and referring expression comprehension.

The vision encoder has 24 layers with 1,024 hidden size and 4,096 FFN intermediate size. The multimodal large language model component is a 24-layer MAGNETO Transformer (Wang et al., 2022a; Ma et al., 2022) with 2,048 hidden dimensions, 32 attention heads, and 8,192 FFN intermediate size. The total number of trainable parameters amounts to approximately 1.6B. The image resolution is set to 224×224 and the patch size is 14×14. To discretize the continuous coordinates, we divide the width and height of the image into 32 equally sized bins, with each bin encompassing an area of 7×7 pixels. A total of 32×32 location tokens are added to the vocabulary. KOSMOS-2 uses the weights of KOSMOS-1 for initialization, the newly added 32×32 word embeddings of location tokens are initialized randomly. We update all the parameters during training and instruction tuning.

**Instruction Tuning**    After model training, instruct tuning is used to better align KOSMOS-2 with human instructions. We combine vision-language instruction dataset (*i.e.*, LLaVA-Instruct (Liu et al., 2023a)) and language-only instruction datasets (*i.e.*, Unnatural Instructions (Honovich et al., 2022) and FLANv2 (Longpre et al., 2023)) with the training data to tune the model. In addition, we construct grounded instruction data by utilizing the pairs of bounding boxes and text spans in GRIT. Given an expression-bounding-box pair, we use "<p> *expression* </p>" as the input instruction, and prompt the model to generate the corresponding location tokens of the bounding boxes. We also use the prompt like "<p> *It* </p><box><loc_{tl}><loc_{br}></box> *is*" to ask the model to generate expressions according to its bounding boxes. More templates are included in Appendix C.2.

## 4    EVALUATION

KOSMOS-2 is initially assessed on multimodal grounding (Sec. 4.1) and multimodal referring (Sec. 4.2) tasks to evaluate its new capabilities, while also being tested on perception-language (Sec. 4.3) and language tasks (Sec. 4.4) to examine conventional MLLM abilities. As mentioned in Sec. 3.2, the grounding capability allows new applications to emerge for KOSMOS-2. More details can be found in Appendix A.

### 4.1    MULTIMODAL GROUNDING

To evaluate the ability of multimodal grounding, we use a generative fashion to test KOSMOS-2 on grounding tasks including phrase grounding and referring expression comprehension. The former requires the model to predict a set of bounding boxes based on one or more given phrases that maybe interrelated within a single caption. The latter encourages the model to locate the object described in a text referring expression within a given image.

For both phrase grounding and referring expression comprehension tasks, KOSMOS-2 generates location tokens which are then converted to bounding boxes for evaluation. The input format is "<image> Image Embedding </image><grounding>...", where "<grounding>" is used to prompt the model to generate locations tokens in its response.

| Model | Zero-shot | Val Split | | | Test Split | | |
|---|---|---|---|---|---|---|---|
| | | R@1 | R@5 | R@10 | R@1 | R@5 | R@10 |
| VisualBert (Li et al., 2019) | ✗ | 70.4 | 84.5 | 86.3 | 71.3 | 85.0 | 86.5 |
| MDETR (Kamath et al., 2021) | ✗ | 83.6 | 93.4 | 95.1 | 84.3 | 93.9 | 95.8 |
| GLIP (Li et al., 2022b) | ✗ | 86.7 | 96.4 | 97.9 | 87.1 | 96.9 | 98.1 |
| FIBER (Dou et al., 2022) | ✗ | 87.1 | 96.1 | 97.4 | 87.4 | 96.4 | 97.6 |
| GRILL (Jin et al., 2023) | ✓ | - | - | - | 18.9 | 53.4 | 70.3 |
| KOSMOS-2 | ✓ | 77.8 | 79.2 | 79.3 | 78.7 | 80.1 | 80.1 |

Table 1: Phrase grounding results on Flickr30k Entities. We report the R@1, R@5, and R@10 metrics, where R@1/5/10 means calculating the recall using the top 1/5/10 generated bounding boxes.

### 4.1.1 PHRASE GROUNDING

We evaluate phrase grounding task on Flickr30k Entities (Plummer et al., 2015) val and test splits. To reduce ambiguity, we do not prompt the model with individual phrases; instead, we use the current phrase along with the preceding words as input where preceding words serve as context: " ... <p> {*phrase*} </p>". For the example shown in Figure 3(1), the model needs to predict the locations of phrases "*A man*", "*a blue hard hat*", "*orange safety vest*" and "*an intersection*" in the caption "*A man in a blue hard hat and orange safety vest stands in an intersection.*". To generate location tokens for the phrase "*A man*" that is the beginning of the caption, the prompt is "<p>*A man*</p>". For phrase "*orange safety vest*", the prompt is "*A man in a blue hard hat and* <p>*orange safety vest*</p>". When there are multiple persons in the image, the context "*A man in a blue hard hat and*" explicitly helps the model locate the object to reduce ambiguity.

We convert location tokens in "<box>...</box>" from the model's response into bounding boxes. A generated bounding box is correct if its intersection over union (IoU) with the ground-truth bounding box is greater than 0.5. If KOSMOS-2 generates a location sequence that can not be converted correctly (*e.g.*, "<box><loc$_{tl}$></box>"), we treat it as a negative sample. We use the ANY-BOX protocol in MDETR (Kamath et al., 2021) and report the R@1, R@5, and R@10 metrics, where R@1/5/10 means calculating the recall using the top 1/5/10 generated bounding boxes. If there are fewer than 5 or 10 bounding boxes generated by KOSMOS-2, we use all available bounding boxes.

**Results** Table 1 presents results on Flickr30k Entities (Plummer et al., 2015) val and test splits. KOSMOS-2 outperforms GRILL (Jin et al., 2023), which relies on an attached detector, by a large margin under zero-shot setting. Moreover, our model outperforms the finetuned VisualBert (Li et al., 2019) model by 7.4% R@1 on both val and test splits. Compared with other models, KOSMOS-2 does not involve prior designs (*e.g.*, object queries or proposals), leading to similar results among R@1, R@5, and R@10. These results demonstrate that KOSMOS-2 can generate high-quality locations without the need for post-processing redundant locations.

### 4.1.2 REFERRING EXPRESSION COMPREHENSION

The model is tested using three well-established datasets: RefCOCO (Yu et al., 2016), RefCOCO+ (Yu et al., 2016) and RefCOCOg (Mao et al., 2015). Both RefCOCO and RefCOCO+ were generated through a two-player game while RefCOCO+ is designed to exclude spatial relations. RefCOCOg incorporates spatial relations and features longer expressions. Different from phrase grounding on Flickr30k entities, we measure this task by using referring expression as the input: "<image> Image Embedding </image><grounding> <p> *referring expression* </p>". For the example in Figure 3(2), the input sequence is "<p>*A man in a blue hard hat and orange safety vest*</p>". The predicted bounding box is correct if its IoU with the ground-truth bounding box is greater than 0.5. The failed decoded sequence is treated as a negative sample. Regardless of whether the model's response contains one or multiple bounding boxes, we only use the first generated bounding box to measure the accuracy.

**Results** Table 2 reports referring comprehension results on RefCOCO, RefCOCO+ and RefCOCOg. KOSMOS-2 also obtains promising zero-shot performance on the comprehension task, significantly

| Model | Zero-shot | RefCOCO | | | RefCOCO+ | | | RefCOCOg | |
|---|---|---|---|---|---|---|---|---|---|
| | | val | testA | testB | val | testA | testB | val | test |
| UNITER (Chen et al., 2019) | ✗ | 81.41 | 87.04 | 74.17 | 75.90 | 81.45 | 66.70 | 74.86 | 75.77 |
| MDETR (Kamath et al., 2021) | ✗ | 87.51 | 90.40 | 82.67 | 81.13 | 85.52 | 72.96 | 83.35 | 83.31 |
| OFA (Wang et al., 2022c) | ✗ | 90.05 | 92.93 | 85.26 | 84.49 | 90.10 | 77.77 | 84.54 | 85.20 |
| FIBER (Dou et al., 2022) | ✗ | 90.68 | 92.59 | 87.26 | 85.74 | 90.13 | 79.38 | 87.11 | 87.32 |
| VisionLLM (Wang et al., 2023) | ✗ | 86.70 | - | - | - | - | - | - | - |
| GRILL (Jin et al., 2023) | ✓ | - | - | - | - | - | - | - | 47.50 |
| KOSMOS-2 | ✓ | 52.32 | 57.42 | 47.26 | 45.48 | 50.73 | 42.24 | 60.57 | 61.65 |

Table 2: Accuracy of referring expression comprehension.

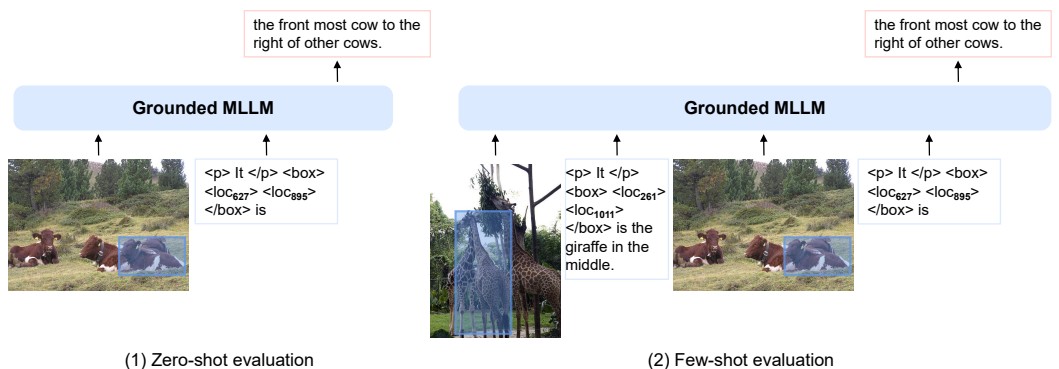

(1) Zero-shot evaluation                 (2) Few-shot evaluation

Figure 4: The input format of referring expression generation evaluation under (1) zero-shot and (2) few-shot settings. The bounding boxes shown in the image are for visualization purposes.

outperforming previous zero-shot models on RefCOCOg benchmark. However, compared to previous well-finetuned works, KOSMOS-2 achieves slightly lower performance on RefCOCO and RefCOCO+ than on RefCOCOg. This discrepancy can be attributed to the data distribution present in RefCOCO and RefCOCO+, where they tend to use a shorter referring expression (*e.g.*, "left bottom") during the two-player game. Hence, one of our future goals is to enhance MLLMs' ability to accurately understand more types of human expressions.

## 4.2 MULTIMODAL REFERRING

In addition to multimodal grounding tasks, we evaluate the model's ability to understand image regions or objects users refer to via inputting bounding boxes. Compared with previous MLLMs that can only refer image regions or objects to the model via detailed text descriptions, directly referring to image regions using its bounding boxes is more effective and reduces ambiguity.

We evaluate the model on the referring expression generation task, which aims to generate unambiguous text descriptions of specific objects or regions within the bounding box. We employ the widely used RefCOCOg dataset (Mao et al., 2015) to evaluate the model's performance under both zero-shot and few-shot settings, showcasing its adaptability in different scenarios.

**Evaluation Setup** The model is tasked with generating an associated text description for an object or region given its location tokens of the bounding boxes (*e.g.*, "`<box><loc`$_{tl}$`><loc`$_{br}$`></box>`"). Benefiting from the unified input format, we use "`<p>` *It* `</p><box><loc`$_{tl}$`><loc`$_{br}$`></box>` *is*" as prompt to encourage the model to predict its text description. Figure 4 (1) and (2) demonstrate the input format for zero-shot and few-shot referring expression generation, respectively. Following previous works, we report results using METEOR and CIDEr metrics. The image resolution is 224×224. Greedy search is used for decoding.

**Results** Table 3 presents the zero-shot and few-shot results of on RefCOCOg. We compare KOSMOS-2 with a finetuned listener-speaker model, which introduces an added reward-based module (SLR). Our model obtains impressive zero-shot performance and even outperforms finetuned SLR by

| Model | Setting | RefCOCOg | |
| --- | --- | --- | --- |
| | | Meteor | CIDEr |
| SLR (Yu et al., 2017) | Finetuning | 15.4 | 59.2 |
| SLR+Rerank (Yu et al., 2017) | Finetuning | 15.9 | 66.2 |
| KOSMOS-2 | Zero-shot | 12.2 | 60.3 |
| | Few-shot ($k = 2$) | 13.8 | 62.2 |
| | Few-shot ($k = 4$) | 14.1 | 62.3 |

Table 3: Results of referring expression generation on RefCOCOg.

| Model | Flickr30k | VQAv2 |
| --- | --- | --- |
| | CIDEr | VQA acc. |
| FewVLM (Jin et al., 2022) | 31.0 | - |
| METALM (Hao et al., 2022) | 43.4 | 41.1 |
| Flamingo-3B (Alayrac et al., 2022) | 60.6 | 49.2 |
| Flamingo-9B (Alayrac et al., 2022) | 61.5 | 51.8 |
| BLIP-2 (Vicuna-13B) (Li et al., 2023b) | 71.6 | **65.0** |
| KOSMOS-1 (Huang et al., 2023) | 67.1 | 51.0 |
| KOSMOS-2 (1.6B) | **80.5** | 51.1 |

Table 4: Zero-shot image captioning results on the Flickr30k test set and zero-shot visual question answering results on the VQAv2 test-dev set.

1.1 CIDEr scores. For the few-shot setting, KOSMOS-2 achieves larger improvements, highlighting its in-context learning ability.

### 4.3 PERCEPTION-LANGUAGE TASKS

In addition to multimodal grounding and referring tasks, we also evaluate KOSMOS-2 on the vision-language tasks. In particular, we perform zero-shot evaluations on two popular tasks, including image captioning and visual question answering. Appendix E provides a more comprehensive comparison on SEED-Bench (Li et al., 2023a) between KOSMOS-2 and recent MLLMs.

For image captioning, we evaluate the model on the widely used Flickr30k *Karpathy split* test set. The beam search algoirthm is used for caption generation, with a beam size of 5. The results are reported using CIDEr (Vedantam et al., 2015) metrics evaluated by COCOEvalCap[1]. The prompt *"An image of"* is used to generate the image description. For visual question-answering, we evaluate zero-shot performance on the test-dev set of VQAv2. Greedy search is used for decoding. We report VQA scores obtained from VQAv2 evaluation server[2]. *"Question: {question} Answer: {answer}"* is used as the prompt for the dataset. The image resolution is 224×224 for both two tasks.

Table 4 displays the zero-shot performance. For image captioning on Flickr30k, KOSMOS-2 with much fewer parameters achieves a remarkable score of 80.5, which significantly outperforms Flamingo-3B (60.6), Flamingo-9B (61.5), and BLIP-2 (Vicuna-13B) (71.6) with large margins. For visual question answering on VQAv2, KOSMOS-2 achieves a VQA accuracy of 51.1, which is on par with KOSMOS-1 (51.0) but lower than BLIP-2 (Vicuna-13B) at 65.0. With more comprehensive functions, *e.g.*, grounding and referring capabilities, KOSMOS-2 demonstrates competitive performance in perception-language tasks.

### 4.4 LANGUAGE TASKS

We evaluate KOSMOS-2 on eight language tasks, including cloze and completion tasks (StoryCloze, HellaSwag), Winograd-style tasks (Winograd, Winogrande), commonsense reasoning (PIQA), and

---

[1]https://github.com/salaniz/pycocoevalcap
[2]https://eval.ai/challenge/830/overview

| Model | Story Cloze | Hella Swag | Winograd | Winogrande | PIQA | BoolQ | CB | COPA |
|---|---|---|---|---|---|---|---|---|
| LLM | 72.9 | 50.4 | 71.6 | 56.7 | 73.2 | 56.4 | 39.3 | 68.0 |
| KOSMOS-1 | 72.1 | 50.0 | 69.8 | 54.8 | 72.9 | 56.4 | 44.6 | 63.0 |
| KOSMOS-2 | 72.0 | 49.4 | 69.1 | 55.6 | 72.9 | 62.0 | 30.4 | 67.0 |

Table 5: Zero-shot performance comparisons of language tasks between KOSMOS-2, KOSMOS-1 and LLM. LLM uses the same text data and training setup to reimplement a language model as KOSMOS-1. For a fair comparison, we report results of KOSMOS-2 and KOSMOS-1 without instruction tuning. Results of KOSMOS-1 and the LLM baseline are from Huang et al., 2023.

three SuperGLUE benchmark (Wang et al., 2019) datasets (BoolQ, CB, and COPA). We report the zero-shot results in Table 5. Compared with KOSMOS-1, KOSMOS-2 achieves similar performance on StoryCloze, HellaSwag, Winograd, Winogrande, and PIQA, experiences a decrease in performance on CB, but shows improvement on BoolQ and COPA. In summary, KOSMOS-2 demonstrates new multimodal grounding and referring capabilities when achieving comparable performance on language tasks, which demonstrates its potential to be a versatile model.

## 5 RELATED WORK

The thriving development of large language models (LLMs, Brown et al., 2020; Chowdhery et al., 2022; Touvron et al., 2023) has paved the way for multimodal large language models (MLLMs, OpenAI, 2023; Alayrac et al., 2022; Wang et al., 2022b; Li et al., 2023b; Huang et al., 2023; Driess et al., 2023; Pan et al., 2023; Lv et al., 2023), which seek to integrate language understanding and reasoning with multimodal perception and comprehension. Flamingo (Alayrac et al., 2022) fuses a pretrained vision encoder and an LLM by introducing gated cross-attention structures, demonstrating impressive multimodal in-context learning capability. KOSMOS-1 (Huang et al., 2023) is another work showing impressive performance under zero/few-shot and multimodal chain-of-thought prompting settings. It is trained from scratch using web-scale multimodal corpora. Recently, instruction-tuning based MLLMs (Liu et al., 2023a; Zhu et al., 2023; Dai et al., 2023; Ye et al., 2023; Gong et al., 2023) endow pretrained LLMs (Touvron et al., 2023; Chiang et al., 2023) multimodal instruction-following capability by constructing high-quality multimodal instruction datasets. Meanwhile, some works are proposed to bridge vision systems with LLMs. VisionLLM (Wang et al., 2023) provides a flexible interaction interface for visual tasks, such as object detection, and segmentation. DetGPT (Pi et al., 2023) combines an MLLM and an extra detector (Liu et al., 2023b) for grounding.

Compared to detectors or grounding models (Chen et al., 2021; Yang et al., 2021; Li et al., 2022b; Liu et al., 2023b), KOSMOS-2 benefits from the advantages of LLMs, such as the ability to comprehend more complex linguistic descriptions and perform superior reasoning. In contrast to existing MLLM methods, KOSMOS-2 incorporates grounding as a foundational capability for MLLMs in various downstream applications, resulting in more informative and comprehensive predictions. Please see Appendix F for more comparisons.

## 6 CONCLUSION

We proposed KOSMOS-2, a multimodal large language modal, that can ground to the visual world. Specifically, we pretrained KOSMOS-2 by augmenting the multimodal corpora used in KOSMOS-1 with GRIT, a large-scale dataset of Grounded Image-Text pairs, which is created by extracting and associating noun phrases and referring expressions in the caption to the objects or regions in the scene. KOSMOS-2 enabled new capabilities of perceiving image regions and grounding text output to the visual world, which makes grounding as a foundation capability of MLLMs in many downstream applications. Extensive experiments demonstrated that KOSMOS-2 exhibited competitive performance in language tasks, while achieving impressive results in vision-language tasks, grounding tasks, and referring tasks. KOSMOS-2 sheds a light on the big convergence of language, multimodal perception, multimodal grounding, and multimodal referring.

## ACKNOWLEDGEMENT

Some example figures are from the WHOOPS corpus (Bitton-Guetta et al., 2023). We would like to acknowledge ydshieh from HuggingFace for both the online demo and the integration of our work into the HuggingFace's transformers. We would also like to acknowledge the efforts of the NVIDIA team for integrating KOSMOS-2 into the NVIDIA foundation model endpoints.

## ETHICS STATEMENT

The model presented in this paper is intended for academic and research purposes. The utilization of the model to create unsuitable material is strictly forbidden and not endorsed by this work. The accountability for any improper or unacceptable application of the model rests exclusively with the individuals who generated such content. We also put Microsoft AI Principles[3] into practice when developing the models.

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

## A    EXAMPLES OF KOSMOS-2

We evaluate KOSMOS-2 on a variety of tasks, including multimodal grounding, multimodal referring, vision-language, and language tasks, as discussed in Section 4. To provide a more intuitive understanding of the model's capabilities, we have included several visualizations in this section. Figure 5 (1) illustrates an example of multimodal grounding, while Figure 5 (4-6) and Figure 9 (2) showcase multimodal referring via bounding boxes.

However, as previously mentioned, the grounding capability of KOSMOS-2 enables a range of new applications to emerge. For instance, Figure 6 highlights the potential of the multimodal referring capability for enhancing human-AI interaction in visual dialogue. In Figure 7, our approach exhibits its in-context learning ability for fine-grained object detection, utilizing both text and image descriptions. Examples of grounded visual question answering can be seen in Figure 5 (2-3) and Figure 8 (1), while Figure 5 (7) and Figure 9 demonstrate grounded detailed image captioning.

Image credits: We would like to express our gratitude for the images sourced from the WHOOPS corpus (Bitton-Guetta et al., 2023), SA-1B (Kirillov et al., 2023), and MS COCO (Lin et al., 2014).

## B    TRAINING HYPERPARAMETERS

The hyperparameters of KOSMOS-2 are listed in Table 6, while the instruction tuning hyperparameters are listed in Table 7.

| Hyperparameters | |
| --- | --- |
| Image embedding number | 64 |
| Location tokens | 1,024 |
| Training steps | 60,000 |
| Warmup steps | 375 |
| Optimizer | AdamW |
| Learning rate | 2e-4 |
| Learning rate decay | Linear |
| Adam $\beta$ | (0.9, 0.98) |
| Weight decay | 0.01 |
| Batch size of text corpora | 93 |
| Batch size of original image-caption pairs | 1,117 |
| Batch size of grounded image-text pairs | 1,117 |
| Batch size of interleaved data | 47 |

Table 6: Hyperparameters of KOSMOS-2

| Hyperparameters | |
| --- | --- |
| Training steps | 10,000 |
| Warmup steps | 375 |
| Learning rate | 1e-5 |
| Batch size of language instruction data | 117 |
| Batch size of vision-language instruction data | 351 |
| Batch size of grounded image-text pairs & grounded instruction data | 1404 |
| Batch size of text corpora | 30 |
| Batch size of interleaved data | 15 |

Table 7: Instruction tuning hyperparameters of KOSMOS-2

## C    GRIT

### C.1    COMPARISON WITH OTHER DATASETS

We compare the created GRIT with existing publicly accessible visual grounding datasets in Table 8. One can see that our GRIT dataset significantly outperforms other existing visual grounding datasets in terms of the number of images, objects, and text spans. With 90,614,680 images, 137,349,210 objects, and 114,978,233 text spans, GRIT is a considerably larger dataset than any of its counterparts, such as Flickr Entities (Plummer et al., 2015), RefCOCOg (Mao et al., 2015), RefCOCO (Yu et al., 2016), RefCOCO+ (Yu et al., 2016), and Visual Genome (Krishna et al., 2016).

In addition, the data in GRIT is more diverse: regarding the text description, GRIT encompasses both referring expressions and noun chunks. Each referring expression or noun chunk could be associated with one or more bounding boxes. More importantly, the image-caption pairs utilized in the GRIT dataset are derived from large-scale web datasets (*e.g.*, LAION-2B (Schuhmann et al., 2022) and COYO-700M (Byeon et al., 2022)), which implies a rich variety of content covering various scenes and topics. This diversity contributes to providing a more comprehensive and realistic setting for visual grounding tasks, ultimately enhancing the generalization capabilities of models.

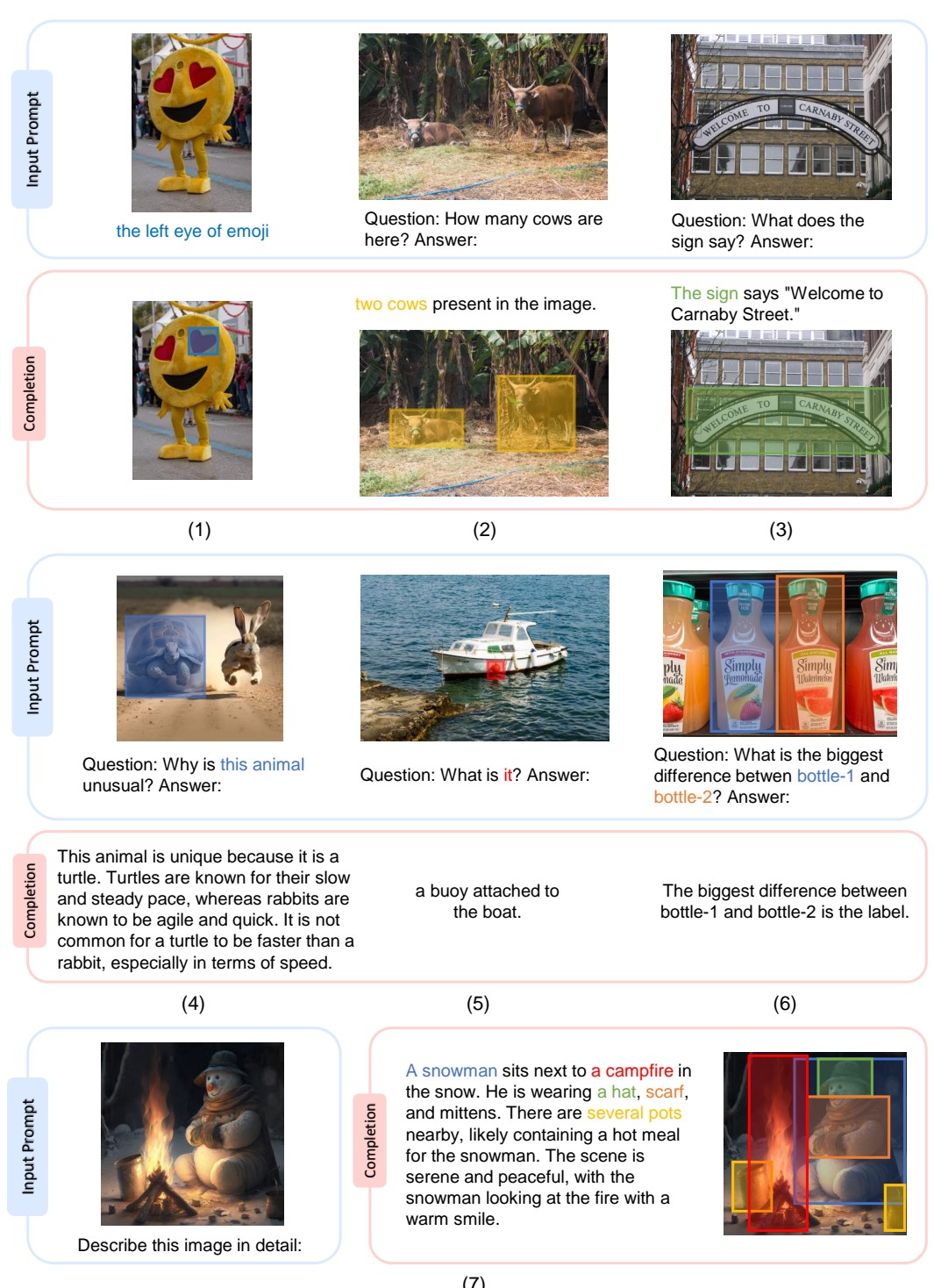

Figure 5: Selected examples generated by KOSMOS-2. The examples include (1) visual grounding, (2)-(3) grounded question answering, (4)-(6) multimodal referring via bounding boxes, and (7) grounded image captioning. (Best viewed in color)

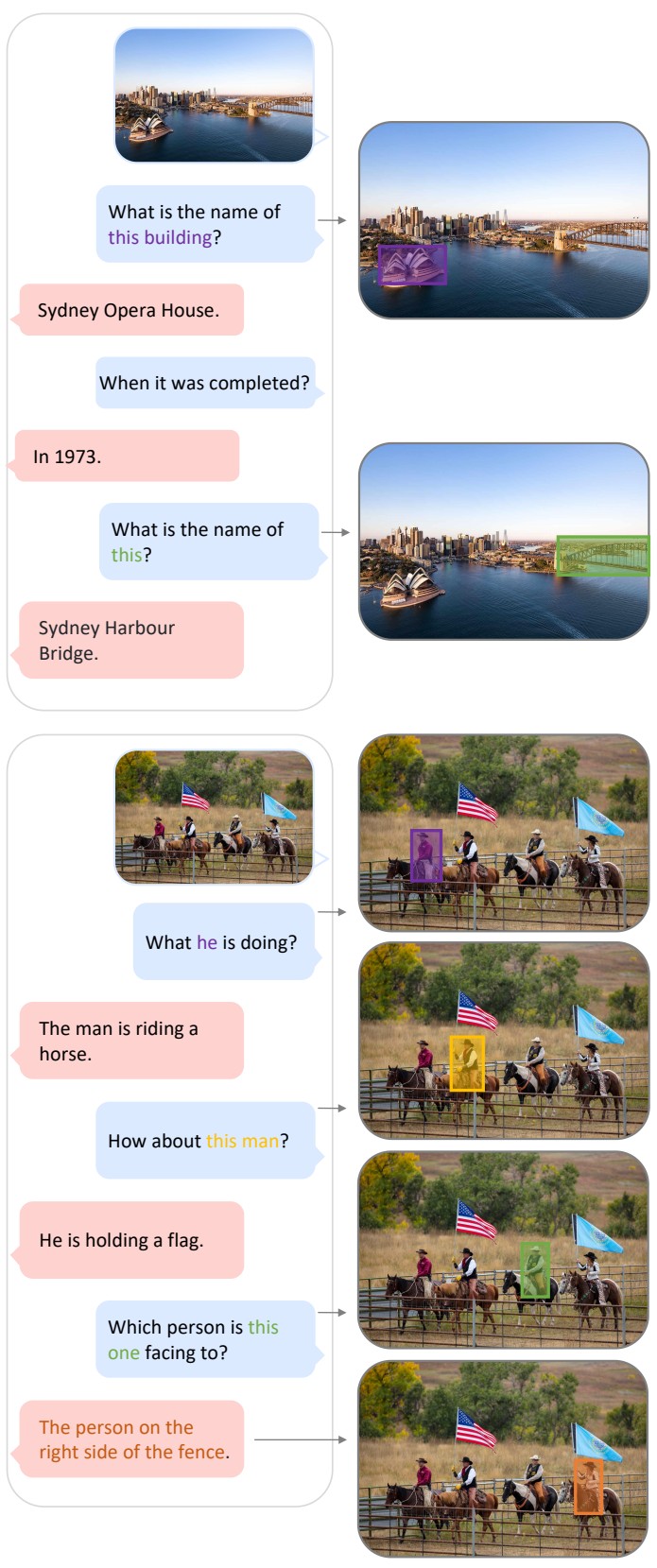

Figure 6: Examples of visual dialogue generated by KOSMOS-2. (Best viewed in color)

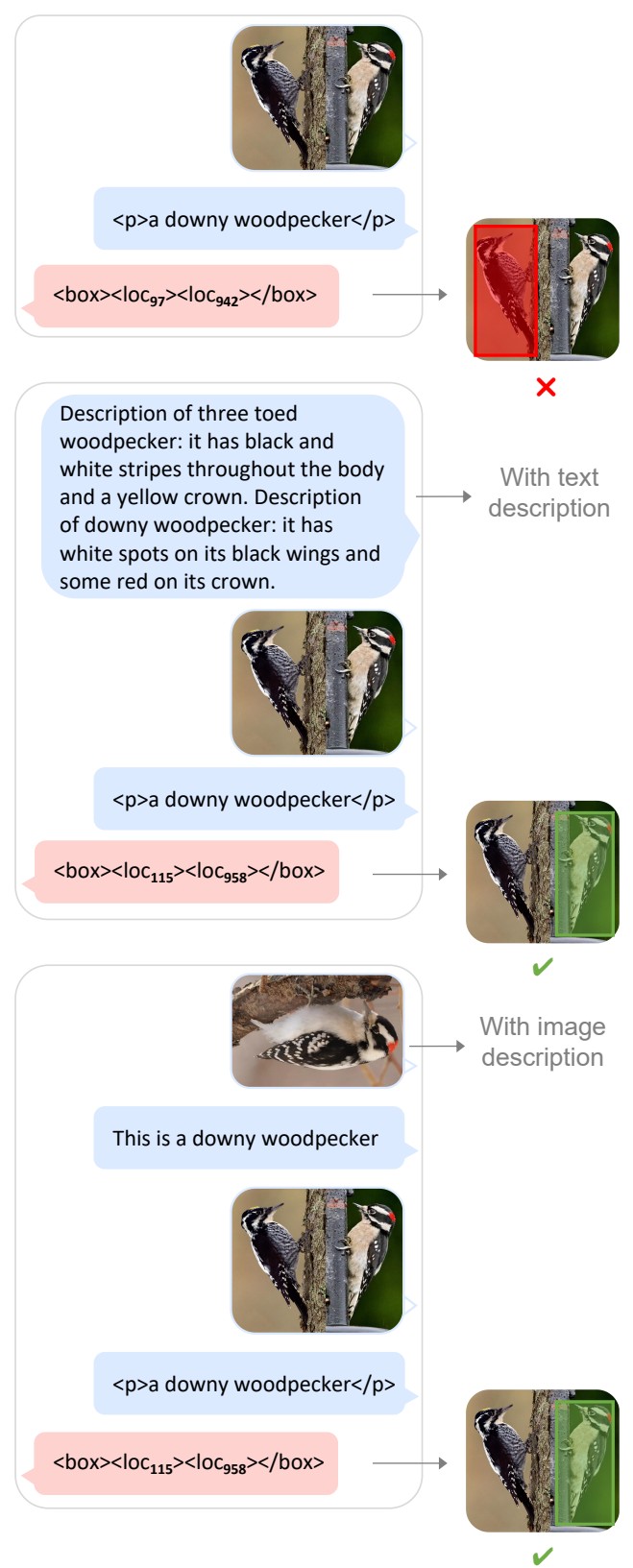

Figure 7: Examples of object detection with multimodal descriptions from KOSMOS-2.

(1) Grounded question answering

(2) Multimodal referring via bounding boxes

Figure 8: Examples generated by KOSMOS-2. (Best viewed in color)

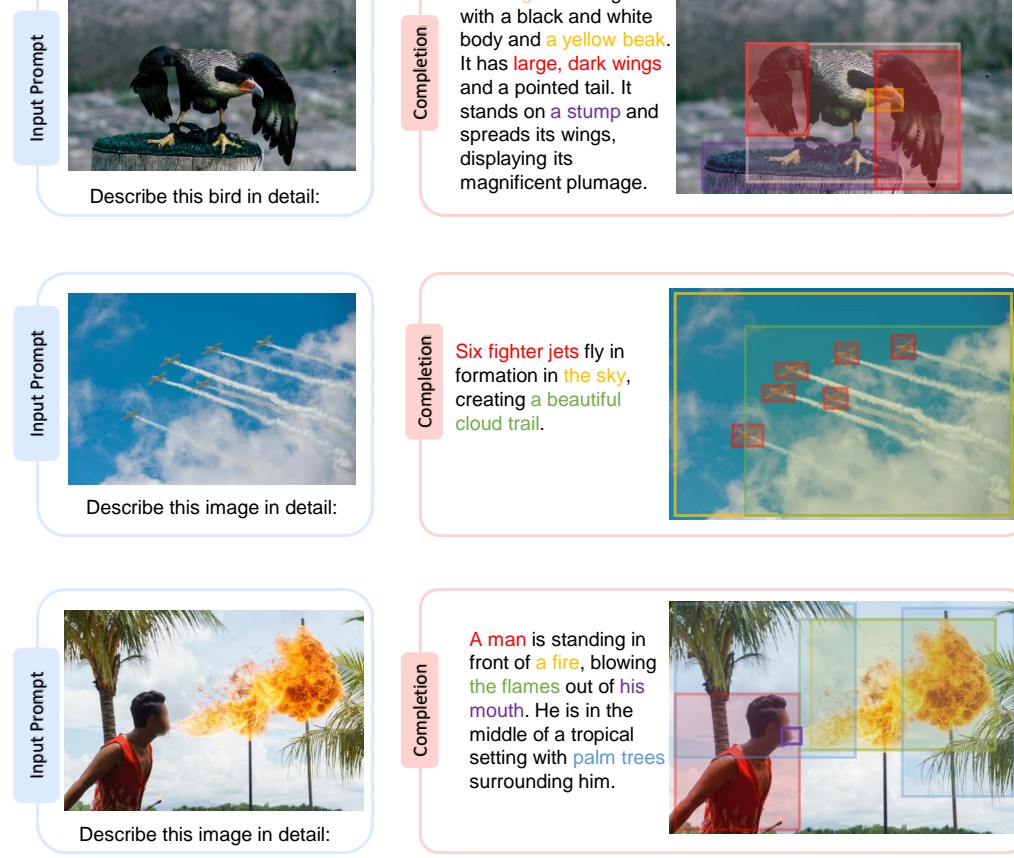

Figure 9: Examples of grounded image captioning generated by KOSMOS-2. (Best viewed in color)

| Dataset | Images | Objects | Text Spans | Avg Expression Length |
|---|---|---|---|---|
| Flickr Entities (Plummer et al., 2015) | 31,783 | 275,775 | 513,644 | - |
| RefCOCOg (Mao et al., 2015) | 26,711 | 54,822 | 85,474 | 8.43 |
| RefCOCO (Yu et al., 2016) | 19,994 | 50,000 | 142,209 | 3.61 |
| RefCOCO+ (Yu et al., 2016) | 19,992 | 49,856 | 141,564 | 3.53 |
| Visual Genome (Krishna et al., 2016) | 108,077 | 4,102,818 | - | - |
| **GRIT (Ours)** | 90,614,680 | 137,349,210 | 114,978,233 | 4.7 |

Table 8: Comparison GRIT with existing visual grounding datasets.

## C.2 TEMPLATES FOR GROUNDED INSTRUCTION DATA

Table 9 presents the instruction templates of expression generation based on its associated bounding boxes during instruction tuning.

## C.3 EXAMPLES OF GRIT

We present some examples of the GRIT corpus in Figures 10,11,12,13. The grounded image-text pairs span over various domains and contain different numbers of objects.

> 1. "What is \<p\> it \</p\>\<box\>\<loc$_{tl}$\>\<loc$_{br}$\>\</box\>? It is {*expression*}."
> 2. "What is \<p\> this \</p\>\<box\>\<loc$_{tl}$\>\<loc$_{br}$\>\</box\>? This is {*expression*}."
> 3. "Describe \<p\> this object \</p\>\<box\>\<loc$_{tl}$\>\<loc$_{br}$\>\</box\>. This object is {*expression*}."
> 4. "\<p\> It \</p\>\<box\>\<loc$_{tl}$\>\<loc$_{br}$\>\</box\> is {*expression*}."
> 5. "\<p\> This \</p\>\<box\>\<loc$_{tl}$\>\<loc$_{br}$\>\</box\> is {*expression*}."
> 6. "\<p\> The object \</p\>\<box\>\<loc$_{tl}$\>\<loc$_{br}$\>\</box\> is {*expression*}."

Table 9: Instruction templates used for expression generation.

| Ablation Settings | RefCOCO | | | RefCOCO+ | | | Flickr30k | | RefCOCOg | |
|---|---|---|---|---|---|---|---|---|---|---|
| | val | testA | testB | val | testA | testB | val | test | val | test |
| Baseline | 52.3 | 57.4 | 47.3 | 45.5 | 50.7 | 42.2 | 77.8 | 78.7 | 60.6 | 61.7 |
| - Instruction tuning | 49.2 | 56.3 | 43.8 | 45.1 | 50.1 | 40.6 | 76.7 | 77.5 | 59.6 | 60.0 |
| - Expression data | 41.4 | 45.1 | 38.1 | 35.7 | 37.1 | 34.3 | 78.5 | 79.0 | 29.2 | 28.3 |

Table 10: Ablation of linguistic description type of objects. Zero-shot Recall@1 metric is tested on Flick30k while zero-shot accuracy protocol is reported on RefCOCO/+/g.

## D    ABLATION STUDY

In Section 2, a pipeline is proposed for constructing web-scale grounded image-text pairs in two steps: generating noun-chunk-bounding-box pairs and producing referring-expression-bounding-box pairs. This process results in two types of text descriptions: noun-chunks and referring expressions. To assess the impact of these two description types on the final visual grounding performance, ablation studies are conducted. It is found that using referring expression data alone is insufficient for training the model. Therefore, a mixture of the two types of data (text spans) is used for training, which also serves as the default setting.

From Table 10, it can be observed that the performance experiences a slight degradation when the instruction tuning phase is disabled. This can be attributed to our utilization of referring expression data to enhance the multimodal referring capability during the instruction tuning phase. Upon further removal of the referring expression data (i.e., disabling Step 2 in Figure 2), there is a significant decline in the results on referring expression comprehensive benchmarks. This highlights the effectiveness of the construction pipeline we proposed in Section 2.

## E    EVALUATION ON SEED-BENCH

Existing benchmarks for MLLMs are limited by inadequate qualitative examples and unsuitable assessments for open-form output. To address, SEED-Bench (Li et al., 2023a) was proposed as a superior benchmark, which consists of 19K multiple choice questions with accurate human annotations and covers 12 evaluation dimensions across both image and video modalities. This comprehensive and objective benchmark enables precise and in-depth evaluation of MLLMs. On SEED-Bench, we compare KOSMOS-2 with popular MLLMs including MiniGPT4 (Zhu et al., 2023), LLaVA (Liu et al., 2023a), BLIP-2 (Li et al., 2023b), InstructBLIP (Dai et al., 2023), MultiModal-GPT (Gong et al., 2023), mPLUG-Owl (Ye et al., 2023), VideoChat (Li et al., 2023c) and Video-ChatGPT (Maaz et al., 2023), Table 11.

As shwon in Table 11, KOSMOS-2 demonstrates remarkable performance with significantly fewer parameters than the counterparts. Although it is marginally outperformed by InstructBLIP (Dai et al., 2023), KOSMOS-2 surpasses the other models in the comparison. Specifically, KOSMOS-2 achieves the best results in scene understanding, instance location, instance interaction, visual reasoning, and action recognition tasks.

One noteworthy observation is that KOSMOS-2 excels in several instance-level tasks (*e.g.*, instance location, instance interaction), indicating that its grounding capability is crucial for fine-grained

Table 11: Performance comparison on SEED-Bench (Li et al., 2023a). $\mathcal{T}_1$ to $\mathcal{T}_{12}$ represent various tasks in the fields of Image and video understanding and reasoning: $\mathcal{T}_1$ - Scene Understanding, $\mathcal{T}_2$ - Instance Identity, $\mathcal{T}_3$ - Instance Attribute, $\mathcal{T}_4$ - Instance Location, $\mathcal{T}_5$ - Instance Counting, $\mathcal{T}_6$ - Spatial Relation, $\mathcal{T}_7$ - Instance Interaction, $\mathcal{T}_8$ - Visual Reasoning, $\mathcal{T}_9$ - Text Recognition, $\mathcal{T}_{10}$ - Action Recognition, $\mathcal{T}_{11}$ - Action Prediction, and $\mathcal{T}_{12}$ - Procedure Understanding. Furthermore, $\mathcal{T}_I$ represents the average performance across the first nine image-based tasks, $\mathcal{T}_V$ signifies the average performance on the last three video-related tasks, and $\mathcal{T}_{All}$ indicates the mean performance over all twelve tasks.

| Model | Language Model | Performance on 12 tasks | | | | | | | | | | | | | | |
| --- | --- | --- | --- | --- | --- | --- | --- | --- | --- | --- | --- | --- | --- | --- | --- | --- |
| | | $\mathcal{T}_1$ | $\mathcal{T}_2$ | $\mathcal{T}_3$ | $\mathcal{T}_4$ | $\mathcal{T}_5$ | $\mathcal{T}_6$ | $\mathcal{T}_7$ | $\mathcal{T}_8$ | $\mathcal{T}_9$ | $\mathcal{T}_I$ | $\mathcal{T}_{10}$ | $\mathcal{T}_{11}$ | $\mathcal{T}_{12}$ | $\mathcal{T}_V$ | $\mathcal{T}_{All}$ |
| *Large Language Models* | | | | | | | | | | | | | | | | |
| LLaMA | LLaMA-7B | 26.3 | 27.4 | 26.2 | 28.3 | 25.1 | 28.8 | 19.2 | 37.0 | 9.0 | 26.6 | 33.0 | 23.1 | 26.2 | 27.3 | 26.8 |
| Vicuna | Vicuna-7B | 23.4 | 30.7 | 29.7 | 30.9 | 30.8 | 28.6 | 29.8 | 18.5 | 13.4 | 28.2 | 27.3 | 34.5 | 23.8 | 29.5 | 28.5 |
| *Multimodal Large Language Models (Image)* | | | | | | | | | | | | | | | | |
| MultiModal-GPT | LLaMA-7B | 43.6 | 37.9 | 31.5 | 30.8 | 27.3 | 30.1 | 29.9 | 51.4 | 18.8 | 34.5 | 36.9 | 25.8 | 24.0 | 29.2 | 33.2 |
| LLaVA | LLaMA-7B | 42.7 | 34.9 | 33.5 | 28.4 | 41.9 | 30.8 | 27.8 | 46.8 | 27.7 | 37.0 | 29.7 | 21.4 | 19.1 | 23.8 | 33.5 |
| mPLUG-Owl | LLaMA-7B | 49.7 | 45.3 | 32.5 | 36.7 | 27.3 | 32.7 | 44.3 | 54.7 | 28.8 | 37.9 | 26.7 | 17.9 | 26.5 | 23.0 | 34.0 |
| MiniGPT-4 | Vicuna-7B | 56.3 | 49.2 | 45.8 | 37.9 | 45.3 | 32.6 | 47.4 | 57.1 | 11.8 | 47.4 | 38.2 | 24.5 | **27.1** | 29.9 | 42.8 |
| BLIP-2 | Flan-T5-XL | 59.1 | 53.9 | 49.2 | 42.3 | 43.2 | 36.7 | **55.7** | 45.6 | 25.9 | 49.7 | 32.6 | 47.5 | 24.0 | 36.7 | 46.4 |
| InstructBLIP | Vicuna-7B | 60.2 | **58.9** | **65.6** | 43.6 | **57.2** | **40.3** | 52.6 | 47.7 | **43.5** | **58.8** | 34.5 | **49.6** | 23.1 | **38.1** | **53.4** |
| KOSMOS-2(ours) | Decoder-only 1.3B | **63.4** | 57.1 | 58.5 | **44.0** | 41.4 | 37.9 | **55.7** | **60.7** | 25.9 | 54.4 | **41.3** | 40.4 | 27.0 | 37.5 | 50.0 |
| *Multimodal Large Language Models (Video)* | | | | | | | | | | | | | | | | |
| Video-ChatGPT | LLaMA-7B | 37.2 | 31.4 | 33.2 | 28.4 | 35.5 | 29.5 | 23.7 | 42.3 | 25.9 | 33.9 | 27.6 | 21.3 | 21.1 | 23.5 | 31.2 |
| VideoChat | Vicuna-7B | 47.1 | 43.8 | 34.9 | 40.0 | 32.8 | 34.6 | 42.3 | 50.5 | 17.7 | 39.0 | 34.9 | 36.4 | 27.3 | 33.7 | 37.6 |

understanding and reasoning. This highlights the superiority of KOSMOS-2 in handling complex tasks that require a deeper understanding of the underlying data.

# F    DISCUSSION WITH MORE METHODS

In the field of Vision-Language Models (VLMs), several innovative approaches have been proposed, emphasizing the incorporation of spatial information or object detection in pretraining. OFA (Wang et al., 2022c) targets unifying various cross and mono modal tasks including image generation, visual grounding, image captioning, image classification, language modeling, etc., and has achieved impressive performances, by utilizing the concept of bounding boxes as tokens from Pix2Seq (Chen et al., 2021). PEVL (Yao et al., 2022) also processes spatial positions as discrete tokens (Chen et al., 2021) and integrated them with language tokens in a unified masked language modeling framework. X-VLM (Zeng et al., 2021) presents an approach that concentrates on multi-grained pretraining. It leverages bounding boxes to glean region-level visual features, aligning them with fine-grained text descriptions through contrastive learning. GLIPv2 (Zhang et al., 2022) model streamlines the process by merging localization pretraining with vision-language pretraining. It employs three tasks: phrase grounding, region-word contrastive learning, and masked language modeling. GRILL (Jin et al., 2023) leverages object-text alignments for learning object grounding and localization, facilitating task transferability, demonstrating adaptability across various tasks such as visual question answering, captioning, and grounding tasks, with zero or few training instances.

Unlike these approaches, our proposed model, KOSMOS-2, aims to unlock the referring and grounding capabilities of multimodal large language models (MLLMs) by training location and language tokens in an auto-regressive paradigm. KOSMOS-2 not only performs well on conventional vision-language tasks such as image captioning, visual question answering, and visual grounding, but also integrates these capabilities into downstream tasks to enable new applications in an open-ended style, thereby extending the capabilities of MLLMs. As Bugliarello et al. emphasize, teaching VLMs object concepts is essential for effectively learning fine-grained skills (Yao et al., 2022; Zeng et al., 2021; Li et al., 2022a). This perspective provides a plausible explanation for our impressive evaluation results on conventional benchmarks and a comprehensive benchmark for MLLMs.

Previous research (Li et al., 2022a; Lee et al., 2021; Bugliarello et al., 2023a) has underscored the significance of data curation in bolstering performance. In this work, we have taken a different approach by creating the GRIT dataset, specifically designed to unlock new capabilities for MLLMs. The methodology employed in the construction of this dataset is generalizable and could offer valuable insights for the larger research community in creating large-scale, task-specific datasets.

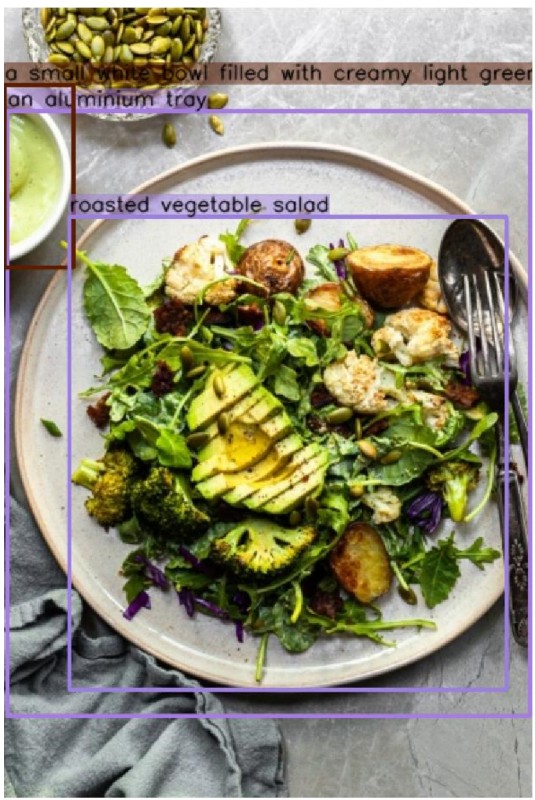

Figure 10: Example from GRIT. Caption: "*A serving of kale and roasted vegetable salad on an aluminium tray served with a small white bowl filed with creamy light green avocado Caesar dressing*".

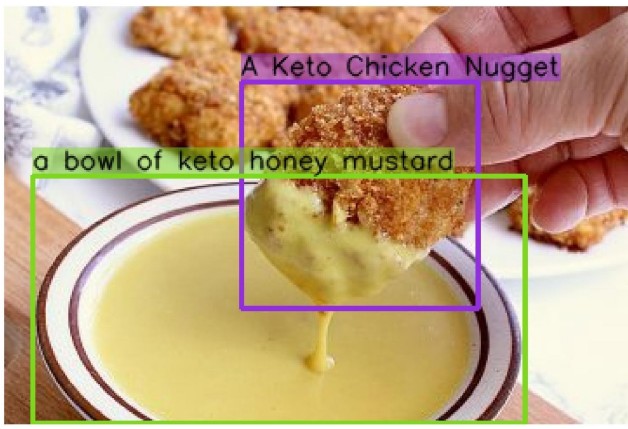

Figure 11: Example from GRIT. Caption: "*A Keto Chicken Nugget being dipped into a bowl of keto honey mustard.*".

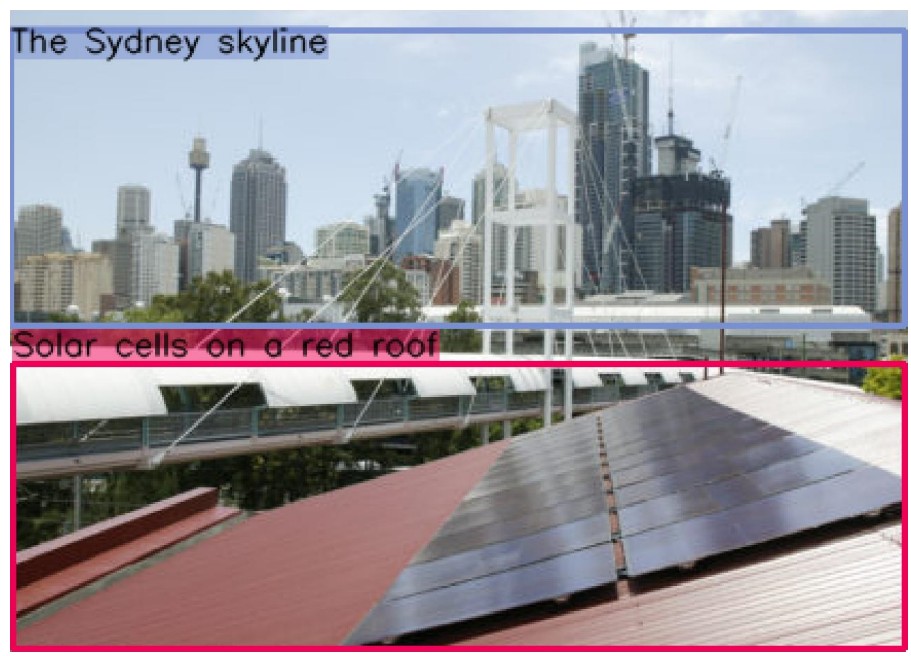

Figure 12: Example from GRIT. Caption: "*Solar cells on a red roof are in the foreground. The Sydney skyline is in the background.*".

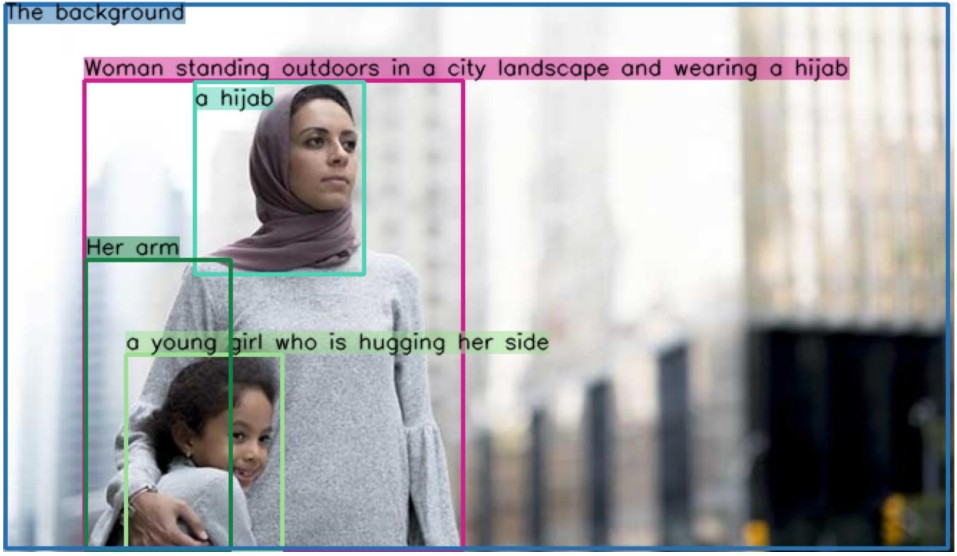

Figure 13: Example from GRIT. Caption: "*Woman standing outdoors in a city landscape and wearing a hijab. Her arm is around a young girl who is hugging her side. The background is blurred.*".

