# OpenReview forum: "Grounding Multimodal Large Language Models to the World"
_ICLR.cc/2024/Conference — ICLR 2024 poster_

### Official Review · Reviewer_873R · 2023-10-26

**Soundness:** 3 good
**Presentation:** 3 good
**Contribution:** 3 good
**Rating:** 6
**Confidence:** 4

**Summary:**

This paper introduces Kosmos-2, a multimodal LLM-based architecture pretrained to perform object localization along standard (image-grounded) text generation. Object localization is also framed as text generation by converting bounding box coordinates to discrete tokens that are added to the language vocabulary. The pretraining data is obtained through an automatic pipeline that extracts phrase–bbox from existing large-scale image caption data. Evaluation on localization-based tasks (phrase grounding, referring expression comprehension/generation) shows that the model acquires strong zero-shot and few-shot learning abilities, while maintaining, while maintaining similar performance on other multimodal and text-only tasks as Kosmos-1 (from which the model is initialized).

**Strengths:**

1. The proposed pipeline to transform image captions into phrase–bbox pairs is interesting, and it holds promise for further development of data manipulation techniques to improve model performance.
2. The proposed model modifications to use the newly generated data consolidate and effectively extend previous approaches that aimed at grounding LMs by adding object location in text.
3. The resulting model obtains strong zero- and few-shot performance on grounding tasks (Sec 4.1 and 4.2), whilst generally maintaining the performance of the model it was initialized from (Kosmos-1)
4. The paper is generally well-written and easy to follow

**Weaknesses:**

1. While generally clear, I do not think I would be able to reproduce the pipeline of Figure 2. The submitted code does not seem to include the steps required to obtain data similar to what is in GrIT. This heavily limits the applicability of the proposed method and future comparisons that aim at improving data manipulation strategies.
2. My main concerns are related to some of the evaluation methodology and claims made in the paper.
- (2a) While results are indeed strong for grounding and referring expressions, I disagree that “Kosmos-2 achieved impressive results on language and vision-language tasks”. The model achieves competitive performance with its baseline.
- (2b) Using Flickr30K as captioning benchmark is suboptimal due to the small number of images (1K) – I encourage the authors to report performance on COCO, which most work uses as benchmark for captioning.
- (2c) While the authors claim that “grounding capability born with KOSMOS-2 enables it to be applied to more downstream tasks, such as grounded image captioning, and grounded visual question answering” no such tasks are tested. An evaluation on a dataset like “VizWiz-VQA-Grounding” [1] would be appreciated towards grounding these claims. Likewise, claims about better action modeling (Conclusion) are unrelated to the proposed model.
- (2d) In Section 4.4, the authors state that “KOSMOS-2 demonstrates new capabilities when achieving comparable performance on language tasks.” This is after showing similar performance on most tasks, worse performance on CV and better performance on BoolQ and COPA. What kind of text-only capabilities are achieved by Kosmos-2? Are there patterns that allow us to pinpoint why the model performs better in these two tasks? And what about performance on CB, the drop is much larger (14.2pp) than any (or even the sum) of the gains on BoolQ and COPA.
3. Another concern that I have is about the unnecessarily claims towards world modeling and AGI. I agree that (better) grounding LLMs to vision goes towards these goals, however, I do not think this should be stressed in the title and abstract (which I invite the authors to consider modifying). While adding additional visual grounding abilities, this approach does not ground “models to the world” in a much more significantly extended way than concurrent work. This can mislead the readers, and given the increased interest in our field from the general public, I believe we researchers should not make claims stronger than needed. The paper proposes good contributions to improve visual grounding, and I think they should be framed as such.
4. The Related Work section needs substantial expansion. Discussion on models like PEVL [2] and X-VLM [3] that pretrain VLMs with object detectors are missing, as well as some of their capabilities [4]. There is also no mention of other approaches towards data curation that can improve model performance, including work like BLIP [5], VSG [6] and other relevant work in NLP [eg, 7].

---
[1] Chen et al. Grounding Answers for Visual Questions Asked by Visually Impaired People. CVPR’22.

[2] Yao et al. PEVL: Position-enhanced Pre-training and Prompt Tuning for Vision-language Models. EMNLP’22.

[3] Zeng et al. Multi-Grained Vision Language Pre-Training: Aligning Texts with Visual Concepts. ICML’22.

[4] Bugliarello et al. Measuring Progress in Fine-grained Vision-and-Language Understanding. ACL’23.

[5] Li et al. BLIP: Bootstrapping Language-Image Pre-training for Unified Vision-Language Understanding and Generation. ICML’22.

[6] Bugliarello et al. Weakly-Supervised Learning of Visual Relations in Multimodal Pretraining. arXiv 2305.2305.

[7] Lee et al. Deduplicating Training Data Makes Language Models Better. ACL’22.

**Questions:**

1. Can you identify any specific abilities of Kosmos-2 that lead to improvement in BoolQ and COPA? And what about systematic failures in CB?
2. In Sec 4.1.2, does the input sequence also include the token “<grounding>” after “</p>”? If not, why? If I understood well, having the token “<grounding>” would better resemble the pretraining GrIT data format.
3. Will you open-source the code to reproduce the pipeline of Figure 2 to produce dataset?

---

> ### Author Response · Authors · 2023-11-20
> **Response by Authors**
>
> We sincerely appreciate your valuable feedback on our paper. The concerns are addressed as follows:
>
> *Q1: Will you open-source the code to reproduce the pipeline of Figure 2 to produce dataset?*
>
> **R1**: Yes, we will open-source the code to reproduce the pipeline of Figure 2 to produce dataset. We have updated the supplementary materials with the code of the pipeline. Please refer to it for more details.
>
> *Q2: Image captioning results on COCO.*
>
> **R2**: Thank you for your suggestions. We have evaluated the model on COCO captioning. The table below shows the zero-shot results on COCO captioning. Kosmos-2 also achieves competitive performance on COCO benchmark.
>
> | Model | CIDEr |
> | --- | --- |
> | VLKD | 58.3 |
> | MetaLM | 82.2 |
> | Flamingo-9B | 79.4 |
> | Kosmos-1 | 84.7 |
> | Kosmos-2 | 90.0 |
>
> *Q3: Results on VizWiz-VQA-Grounding*
>
> **R3**:  We appreciate your suggestion to test Kosmos-2 on the VizWiz-VQA-Grounding dataset. We tried to evaluate the model on the task but find that VizWiz-VQA-Grounding dataset measures performance using mask IoU&mAP, while Kosmos-2 currently only supports bounding box output. We have reached out to the authors of the dataset and were informed that no bounding box metric has been prepared or tested. Given these circumstances, we instead measure the accuracy metric as in RefCOCO. When the bounding rectangle of the ground truth mask is used as the ground truth box, Kosmos-2 achieves 58.9% top-1 accuracy under the zero-shot setting.
>
> *Q4: Results on language tasks*
>
> **R4**: I think our statement may have caused some misunderstanding. Here, "new capabilities" refers to grounding and referring, not language capabilities. We will make it clear in the revised paper. What we want to express more is that while the model has new capabilities of grounding and referring, its previous language abilities have not declined. For the performance improvement on BoolQ and COPA, and failure on CB, we will try to explore the possible patterns.
>
> *Q5: Unnecessarily claims*
>
> **R5**: Thank you for your suggestions, we will try to correct those inappropriate claims
>
> *Q6: Disscussion with more methods*
>
> **R6**: We have added the discussion of these related works into Appendix F of the revised paper.
>
> *Q7: In Sec 4.1.2, does the input sequence also include the token “<grounding>” after “</p>”? If not, why? If I understood well, having the token “<grounding>” would better resemble the pretraining GrIT data format.*
>
> **R7**: Yes, the "<grounding>" token is included in the input sequence, and we place it after the “</image>” token. We will make this clear in the revised version. The "<grounding>" token is employed to signal the model that the subsequent sequence includes text spans and their associated location tokens.

---

> > ### Comment · Reviewer_873R · 2023-11-23
> >
> > Thank you for the clarifications, committing to release the data pipeline, and running extra experiments!

---

### Official Review · Reviewer_Cmrj · 2023-10-31

**Soundness:** 3 good
**Presentation:** 3 good
**Contribution:** 3 good
**Rating:** 8
**Confidence:** 4

**Summary:**

The paper proposes Kosmos-2, a Multimodal Large Language Model (MLLM) that can not only do vision-language tasks based on image inputs, but also enables both inputting and outputing bounding box regions. The authors first introduce how to construct a training dataset of grounded image-text pairs, then illustrate how they enable the model to do grounding and referring by using bounding box representation as a hyperlink. The experimental results demonstrate the new grounding and referring capabilities of the model, and also comparable image understanding ability with other MLLMs.

**Strengths:**

1. The motivation is promising and important. By enabling MLLM to do grounding and referring, it brings more flexibility to the model and enables it to do a lot more fancy things. We can imagen how convenient it will be if users can specify a region when querying a MLLM. The paper makes a first step towards building such a model together with a training dataset.
2. Experimental results show that the model is comparable with previous MLLM on various traditional vision-language tasks, while shows new capabilities for grounding and referring.

**Weaknesses:**

I don't see a big weakness here. One thing is that the proposed model achieved much worse performance than the previous grounding & referrring models. Ideally, with language model's stronger language capabilities, the model should achieve better performance on these tasks. Therefore, there are still rooms here for improvements.

**Questions:**

I am curious about the performance of integrating an object-detection model with Kosmos-2. Say, if we provide the location of the objects for VQA tasks, what performance can the model achieve?

---

> ### Author Response · Authors · 2023-11-20
> **Response by Authors**
>
> We sincerely appreciate your valuable feedback on our paper. The concerns are addressed as follows:
>
> *Q1: Worse performance than the previous grounding & referring models.*
>
> **R1**: We agree that there is a significant room for improvement. This can be achieved by increasing the model size and enhancing the image resolution. Regarding to the performance gap between our model and previous specialist models, we believe there are the following reasons:
>
> 1. Previous specialist models undergo pretraining or finetuning procedures on these specific datasets. In contrast, Kosmos-2 is trained on noisy web-scale data and is evaluated under zero or few-shot settings. This difference in training and testing conditions could partially explain the observed performance gap. We believe that increasing the model size can effectively narrow the gap in performance. We can also add the specific datasets, such as RefCOCO, into our training data to boost the performance. This has been acknowledged to yield good results by models such as QWen-VL[1], CogVLM[2], MiniGPT-v2[3] and Florence-2[4].
> 2. In order to reduce the cost of model training and inference, the input image resolution of our model is 224x224 and we reduce the number of image embeddings to 64 via a resampler before feeding them into the language model. While previous specialist models often employ a much larger image resolution to obtain better performance. We will increase the image resolution and number of image embeddings in future work.
>
> *Q2: I am curious about the performance of integrating an object-detection model with Kosmos-2. Say, if we provide the location of the objects for VQA tasks, what performance can the model achieve?*
>
> **R2**: We tested some VQA examples on the model and provided location information for relevant objects in the questions. We found that for some examples where the object being questioned is not clearly referred to by the text referring expression, providing their location information can help the model get the correct answer. Providing location information can resolve coreference ambiguity. In addition, we also tested the model on the grounded VQA benchmark VizWiz-VQA-Grounding, as suggested by Reviewer 873R, and achieved 58.9% accuracy. The model can locate the image region used to arrive at the answer.
>
> [1] Bai, Jinze et al. “Qwen-VL: A Versatile Vision-Language Model for Understanding, Localization, Text Reading, and Beyond.”
>
> [2] Wang, Weihan et al. “CogVLM: Visual Expert for Pretrained Language Models.”
>
> [3] Chen, Jun et al. “MiniGPT-v2: large language model as a unified interface for vision-language multi-task learning.”
>
> [4] Xiao, Bin et al. “Florence-2: Advancing a Unified Representation for a Variety of Vision Tasks.”

---

### Official Review · Reviewer_n3ax · 2023-11-01

**Soundness:** 3 good
**Presentation:** 3 good
**Contribution:** 2 fair
**Rating:** 6
**Confidence:** 4

**Summary:**

The paper presents a way to allow multimodal language models to refer to and ground objects in images. Grounded text spans are represented as [text span](bounding boxes), and similarly one could use such formats to input and output bounding boxes. A separate grounding model is used then to annotate boxes for web image-text data and Kosmos-2 is pre-trained on such image-grounded text data. Evaluation shows that the model archives zero/few-shot performance comparable to previous supervised grounding / grounded understanding models.

**Strengths:**

- The idea is simple and effective. Using such markdown-like formats allows the model to easily take boxes as input or output grounding boxes. The authors also spend time discussing the details in converting image-text data into such grounded data, including how to extract the correct noun chunk (Section 2), and how to convert boxes into a markdown-like format (Section 3.1), which is welcomed.

- The evaluation shows the model can handle diverse tasks: 1) accepting a phrase and outputting boxes; 2) accepting boxes and outputting descriptions; 3) classical VL tasks such as VQA and language understanding tasks.

**Weaknesses:**

1. Limited methodology innovation

    There is no significant methodology improvement, as outputting bounding boxes as discrete tokens has been extensively studied in prior work such as [1].

    [1]. OFA: Unifying Architectures, Tasks, and Modalities Through a Simple Sequence-to-Sequence Learning Framework. Wang et al., 2022.

2. Ablation study

     While the evaluation is extensive, there is no ablation study or discussion of key design choices or error analysis. Thus it is hard to draw further insights from the experiments. Below I just list a few questions that I have but I would encourage the authors to include ablation study / error analysis.

     a. How much impact does the data cleaning process (Section 2) have on the final performance?

     b. Where does the in-context learning ability come from? Is it solely from interleaved image-text data or are in-context grounded data specifically included?

     c. Can we fine-tune Kosmos on downstream grounding tasks? If so, will the system match previous fine-tuned models?

     d. Few-shot in-context prompting results are shown only for RefCOCOg. Could we also do few-shot prompting for grounding tasks such as Flickr30K?

**Questions:**

Will some of the grounded data (converted from public data) be made public?

---

> ### Author Response · Authors · 2023-11-20
> **Response by Authors**
>
> We sincerely appreciate your valuable feedback on our paper. The concerns are addressed as follows:
>
> *Q1: Limited methodology innovation.*
>
> **R1**: While we acknowledge the contribution of Pix2Seq and other works such as OFA in considering bounding boxes as tokens, we believe our work, Kosmos-2, has its unique positioning and contributions:
> 1. Although both OFA and Kosmos-2 utilize the concept of bounding boxes as tokens from Pix2Seq, the motivations and goals of the two models are different. OFA aims to unify various cross-modal and mono-modal tasks via a unified sequence-to-sequence learning paradigm for pretraining and finetuning, and it has achieved impressive performances on a wide range of benchmarks. In contrast, Kosmos-2 focuses on unlocking the referring and grounding capabilities of MLLMs and integrates these capabilities into downstream applications in an open-ended style.
> 2. We are the first to verify the feasibility of using a simple yet effective 'hyperlink' modeling strategy to equip Multimodal Large Language Models with grounding capabilities while preserving conventional functionalities.
> 3. We also propose a straightforward pipeline for constructing web-scale grounded image-text pairs to enable potential applications for the community. We are delighted to see that our pipeline has been helpful to other works within the field.
>
> *Q2: How much impact does the data cleaning process (Section 2) have on the final performance?*
>
> **R2**: Thank you for your valuable suggestion on the inclusion of an ablation study. We have done some ablation studies and have included the results and analysis in the Appendix D of the revised paper. We summarize the results as follows:
> | Ablation Settings | RefCOCO (val) | RefCOCO (testA) | RefCOCO (testB) | RefCOCO+ (val) | RefCOCO+ (testA) | RefCOCO+ (testB) | RefCOCOg (val) | RefCOCOg (test) | Flickr30k (val) | Flickr30k (test) |
> | --- | --- | --- | --- | --- | --- | --- | --- | --- | --- | --- |
> | Baseline | 52.3 | 57.4 | 47.3 | 45.5 | 50.7 | 42.2 | 60.6 | 61.7 | 77.8 | 78.7 |
> | - Instruction tuning | 49.2 | 56.3 | 43.8 | 45.1 | 50.1 | 40.6 | 59.6 | 60.0 | 76.7  | 77.5 |
> | - Step 2 in Figure 2 | 41.4  | 45.1  | 38.1  | 35.7  | 37.1  | 34.3  | 29.2  | 28.3 | 78.5  | 79.0 |
>
> *Q3: Where does the in-context learning ability come from? Is it solely from interleaved image-text data or are in-context grounded data specifically included? Few-shot in-context prompting results are shown only for RefCOCOg. Could we also do few-shot prompting for grounding tasks such as Flickr30K?*
>
> **R3**: We do not incorporate in-context grounded data, but we are very interested in trying in the future. Determining the precise source of the in-context learning ability is a complex task as it involves several factors such as data distribution, model architecture, and scale. This is indeed a fascinating and significant research topic in the age of LLM. In the context of Kosmos-2, the use of interleaved image-text data and text corpus seems to be related to the in-context learning ability on vision-language tasks.  We believe the few-shot capability is general and not specific to any datasets. We have deliberately avoided using previous annotated data to prevent the model from learning dataset-specific bias. Evaluating on grounding tasks such as Flickr30k should be feasible given the generality of the model's in-context learning ability.
>
> *Q4: Can we fine-tune Kosmos on downstream grounding tasks? If so, will the system match previous fine-tuned models?*
>
> **R4**: Absolutely, Kosmos-2 can be fine-tuned on downstream grounding tasks. However, when it is solely fine-tuned on existing grounding datasets, the language model of Kosmos-2 tends to degenerate into a task layer, similar to the Transformer in Pix2Seq[1]. This is not our intended goal. In fact, some subsequent works such as Qwen-VL[2] and CogVLM[3], which incorporate RefCOCO series datasets into the instruction tuning phase, have demonstrated that the final performance can match that of specialist models.
>
> *Q5: Will some of the grounded data (converted from public data) be made public?*
>
> **R5**: Yes, we plan to make the grounded data, converted from public data, available to the public. We will also release all the associated code and model weights. This will facilitate further research in this field and enable others to reproduce and build upon our work.
>
> [1] Chen, Ting et al. “Pix2seq: A Language Modeling Framework for Object Detection.”
>
> [2] Bai, Jinze et al. “Qwen-VL: A Versatile Vision-Language Model for Understanding, Localization, Text Reading, and Beyond.”
>
> [3] Wang, Weihan et al. “CogVLM: Visual Expert for Pretrained Language Models.”

---

> ### Author Response · Authors · 2023-11-22
> **Response by Authors**
>
> Thanks again for reviewing our paper. We hope that we were able to address your concerns in our response. As the deadline is approaching, please let us know if you have any further questions before the reviewer-author discussion period ends. We are glad to address your further concerns.

---

> > ### Comment · Reviewer_n3ax · 2023-11-22
> >
> > Thanks to the authors for the rebuttal. I am raising the score and I think the paper should be accepted. I will modify my review later to reflect the rebuttal.

---

### Official Review · Reviewer_vJ2h · 2023-11-01

**Soundness:** 3 good
**Presentation:** 3 good
**Contribution:** 3 good
**Rating:** 8
**Confidence:** 4

**Summary:**

The paper introduces "KOSMOS-2", a Multimodal Large Language Model (MLLM) that has the ability to perceive object descriptions, such as bounding boxes, and ground textual content to the visual world. This is achieved by representing specific text spans, like referring expressions and noun phrases, as links in Markdown format, connecting the text to location tokens that denote the object's position. A significant dataset called GRIT, consisting of grounded image-text pairs, is constructed to train the model. This dataset is built using subsets from previous image-text pair datasets, namely LAION-2B and COYO-700M. Experimental results demonstrate KOSMOS-2's leading performance in grounding and referring tasks, as well as competitive results in traditional language and vision-language tasks. This grounding capability allows KOSMOS-2 to be used in various applications, such as grounded image captioning and grounded visual question answering. The study underscores the convergence of language, multimodal perception, and world modeling, a vital progression toward artificial general intelligence.

**Strengths:**

The core innovation is the method to link text spans in image captions to spatial coordinates of the objects or regions in the image. These coordinates are converted into location tokens and attached to the text span, forming a "hyperlink" that connects the image's objects or regions to its caption. When trained, KOSMOS-2 can link these text spans in generated text to specific image regions, resulting in more accurate and comprehensive vision-language predictions. The model can also identify and refer to objects using pronouns, enhancing its reasoning capabilities.

Besides that, the paper is well written and clearly presented.

**Weaknesses:**

The GRIT dataset is trained on Grit together with text to image instructional dataset, and the work claims that the model maintain the conventional multimodal capability. It is important to evaluate the model on more multi understanding tasks to validate the modeling approach. The model is only evaluated on Flickr30k and VQAv2, a more comprehensive evaluation is lacking.

**Questions:**

Is there data cleaning process to make sure that training dataset do not contain any image from evaluation dataset?

For table 5, what's the training cost of LLM, Kosmos-1 and Kosmos-2. If LLM is trained longer and match the gpu hours of kosmos-2, will that change the results in table 5.

---

> ### Author Response · Authors · 2023-11-20
> **Response by Authors**
>
> We sincerely appreciate your valuable feedback on our paper. The concerns are addressed as follows:
>
> *Q1: More comprehensive evaluation.*
>
> **R1**: We agree that a more comprehensive evaluation would provide further validation of our approach. In addition to Flickr30k and VQAv2, we have also evaluated Kosmos-2 on the SEED-Bench (Table 11 in Appendix E of the revised paper), which provides a multi-dimensional evaluation across 12 multimodal tasks. We compare Kosmos-2 with multiple MLLMs, such as LLaVA and BLIP-2. Experimental results demonstrate that Kosmos-2 achieves remarkable performance with significantly fewer parameters. Specifically, our model achieves better performance in tasks of scene understanding, instance location, instance interaction, visual reasoning, and action recognition. We believe these results further underline the effectiveness and versatility of Kosmos-2 in various multimodal tasks. We hope this additional information addresses your concern regarding comprehensive evaluation.
>
> *Q2: Is there data cleaning process to make sure that training dataset do not contain any image from evaluation dataset?*
>
> **R2**: Yes, we have performed the data cleaning. We also remove the LLaVA-Instruct-150k data where images appear in the evaluation dataset.
>
> *Q3: For table 5, what's the training cost of LLM, Kosmos-1 and Kosmos-2. If LLM is trained longer and match the gpu hours of kosmos-2, will that change the results in table 5.*
>
> **R3**: The LLM is trained using the same text corpora and training setup of Kosmos-1. Based on Kosmos-1, we train Kosmos-2 on 256 V100 GPUs for 24 hours. Therefore, the computational cost of Kosmos-2 slightly exceeds that of the LLM, and we also believe that training the LLM for the same GPU hours would yield better results. However, in Table 5, what we want to highlight is that while Kosmos-2 has acquired new grounding and referring capabilities, its language performance has not declined. We demonstrate its potential to be a versatile model.

---

> > ### Comment · Reviewer_vJ2h · 2023-11-23
> > **Responce to Authors**
> >
> > Thanks for your feedback and additional results. I would like to keep my original rating.

---

### Meta-Review · Area_Chair_zgT5 · 2023-12-04

**Metareview:**

The idea of explicitly linking textual input/output to bounding boxes in input images is pretty nice. In its more general form, this kind of hyperlinking is promising. There is increasing interest in vision-capable LLMs, but how to do it right remains largely an open problem (maybe there are some successes closed in companies, with limited release of details, if any). So overall, a nice contribution, that will likely interest researchers, and might even be useful in the form of resources.

**Justification For Why Not Higher Score:**

There are evaluation and presentation issues, especially with use of exciting, but vague and inaccurate term. At times, this can even mislead the reader, especially non expert readers. This issue starts in title, as one of the reviewers explicitly complained about, and continues throughout the paper. What is even more unfortunate is that none of this is required to make the paper more exciting. So the authors simply hurt themselves. I do hope they will fix these issues before the camera ready.

**Justification For Why Not Lower Score:**

Solid contribution that is timely and will interest peers.

---

### Decision · Program_Chairs · 2024-01-16

Accept (poster)